# The Thrombopoietin Receptor Agonist Eltrombopag Inhibits Human Cytomegalovirus Replication Via Iron Chelation

**DOI:** 10.3390/cells9010031

**Published:** 2019-12-20

**Authors:** Jens-Uwe Vogel, Sophie Schmidt, Daniel Schmidt, Florian Rothweiler, Benjamin Koch, Patrick Baer, Holger Rabenau, Detlef Michel, Thomas Stamminger, Martin Michaelis, Jindrich Cinatl

**Affiliations:** 1Institut für Medizinische Virologie, Universitätsklinikum, Goethe-Universität, Paul Ehrlich-Str. 40, 60596 Frankfurt am Main, Germany; ju.vogel@kinderkrebsstiftung-frankfurt.de (J.-U.V.); Sophie.Marion.Schmidt@web.de (S.S.); Daniel.Schmidt-94@web.de (D.S.); f.rothweiler@kinderkrebsstiftung-frankfurt.de (F.R.); rabenau@em.uni-frankfurt.de (H.R.); 2Medizinische Klinik III, Nephrologie, Klinikum der Goethe-Universität, Theodor-Stern-Kai 7, 60590 Frankfurt am Main, Germany; b.koch@med.uni-frankfurt.de (B.K.); p.baer@em.uni-frankfurt.de (P.B.); 3Institut für Virologie, Universitätsklinikum Ulm, Albert-Einstein-Allee 23, 89081 Ulm, Germany; detlef.michel@uniklinik-ulm.de (D.M.); thomas.stamminger@uniklinik-ulm.de (T.S.); 4Industry Biotechnology Centre and School of Biosciences, University of Kent, Canterbury CT2 7NJ, UK

**Keywords:** human cytomegalovirus, antiviral therapy, eltrombopag, thrombopietin receptor agonist, drug resistance, iron chelation

## Abstract

The thrombopoietin receptor agonist eltrombopag was successfully used against human cytomegalovirus (HCMV)-associated thrombocytopenia refractory to immunomodulatory and antiviral drugs. These effects were ascribed to the effects of eltrombopag on megakaryocytes. Here, we tested whether eltrombopag may also exert direct antiviral effects. Therapeutic eltrombopag concentrations inhibited HCMV replication in human fibroblasts and adult mesenchymal stem cells infected with six different virus strains and drug-resistant clinical isolates. Eltrombopag also synergistically increased the anti-HCMV activity of the mainstay drug ganciclovir. Time-of-addition experiments suggested that eltrombopag interfered with HCMV replication after virus entry. Eltrombopag was effective in thrombopoietin receptor-negative cells, and the addition of Fe^3+^ prevented the anti-HCMV effects, indicating that it inhibits HCMV replication via iron chelation. This may be of particular interest for the treatment of cytopenias after hematopoietic stem cell transplantation, as HCMV reactivation is a major reason for transplantation failure. Since therapeutic eltrombopag concentrations are effective against drug-resistant viruses, and synergistically increase the effects of ganciclovir, eltrombopag is also a drug-repurposing candidate for the treatment of therapy-refractory HCMV disease.

## 1. Introduction

Eltrombopag is a thrombopoietin receptor (also known as c-Mpl or MPL) agonist that is used for the treatment of thrombocytopenia, including hepatitis C virus-associated thrombocytopenia [1,2,3]. Its use has also been suggested for the treatment of cytopenias after hematopoietic stem cell transplantations and case reports support its safety and efficacy [4,5,6,7,8,9].

Human cytomegalovirus (HCMV) reactivation and HCMV-associated disease are leading reasons for the failure of hematopoietic stem cell transplantations [10,11,12]. Anti-HCMV drugs, including ganciclovir, cidofovir, and foscarnet, are available, but their use is associated with severe side effects [13]. In particular, the use of ganciclovir (and its prodrug valganciclovir), the mainstay treatment for cytomegalovirus disease, is associated with severe hematological side effects, including thrombocytopenia [14,15,16].

A case report described the use of eltrombopag in an immunocompetent patient who suffered from human cytomegalovirus (HCMV)-associated thrombocytopenia [17]. Immunosuppressive treatment for thrombocytopenia (prednisone, intravenous immunoglobulin, dapsone), in combination with antiviral therapy (ganciclovir/valganciclovir, HCMV hyperimmune globulin), only resulted in a temporary platelet response with subsequent relapse. A change to eltrombopag, intended to increase platelet counts without immunosuppressive therapy, resulted in a durable increase in platelet levels, no evidence of HCMV viraemia, and the resolution of symptoms [17]. The observed effects were attributed to eltrombopag overcoming HCMV-induced suppression of platelet production [17]. However, we hypothesized that direct antiviral effects may also have contributed to the beneficial outcome in the case report of the patient with HCMV-associated thrombocytopenia [17]. Indeed, we found that eltrombopag exerts anti-HCMV effects via iron chelation.

## 2. Materials and Methods

### 2.1. Drugs

Eltrombopag (the orally active ethanolamine salt of eltrombopag olamine) was purchased from Selleck Chemicals (via Absource Diagnostics GmbH, Munich, Germany), deferasirox and ganciclovir from MedChemExpress (via Hycultec, Beutelsbach, Germany), and cidofovir from Cayman Chemical (via Biomol GmbH, Hamburg, Germany).

### 2.2. Cells and Viruses

Primary human foreskin fibroblasts (HFFs) and adipose-derived adult mesenchymal stem cells (ASCs) were cultivated as previously described [18,19].

The wild-type HCMV strain, Hi91, was isolated from the urine of an AIDS patient with HCMV retinitis, as described previously [20]. HCMV strains Davis and Towne were received from the American Type Culture Collection (ATCC) (Manassas, VA, USA). Virus stocks were prepared in HFFs maintained in minimal essential medium (MEM), supplemented with 4% fetal calf serum (FCS). The U1, U59, and U75 are patient isolates, which were isolated as previously described [20,21]. Virus stocks were prepared in HFFs maintained in MEM supplemented with 4% FCS.

Murine cytomegalovirus (Smith strain, catalogue number VR-1399) was obtained from ATCC, and virus stocks were prepared in NIH/3T3 mouse fibroblasts (ATCC) maintained in MEM supplemented with 4% FCS.

DNA isolation, amplification, and sequencing were performed as previously described [21], using established primers [22].

### 2.3. Virus Infectivity Assay

In 96-well microtiter plates, confluent cultures of HFF cells or ASCs were incubated with HCMV at the indicated multiplicities of infection (MOIs). After incubation for 1-h, cells were washed with phosphate-buffered saline (PBS) and incubated in MEM containing 4% FCS and serial dilutions of the indicated substances.

As described previously [18,23], cells producing HCMV-specific antigens were detected 24 h post infection by immunoperoxidase staining, using monoclonal antibodies directed against the UL123-coded 72 kDa immediate early antigen 1 (IEA1) (Mouse Anti CMV IEA, MAB8131, Millipore, Temecula, CA, USA), and 120 h post infection by immunoperoxidase staining, using monoclonal antibodies directed against UL55-encoded late antigen (LA) gB, (kindly provided by K. Radsak, Institut für Virologie, Marburg, Germany), as previously described. Drug concentrations that reduced HCMV antigen expression by 50%, inhibitory concentration (IC_50_), were calculated using Calcusyn (Biosoft, Cambridge, UK).

Effects of eltrombopag on murine cytomegalovirus were determined by visual scoring of cytopathogenic effect (CPE) formation (detected 120 h post infection) in MOI 1-infected murine NIH/3T3 fibroblasts.

### 2.4. Drug Combination Studies

Drugs were combined at equimolar concentrations or used as single agents. Combined effects were determined by staining for HCMV LA. Combination indices (CIs) were calculated at different levels of inhibition (50% inhibition, CI_50_; 75% inhibition, CI_75_; 90% inhibition, CI_90_; 95% inhibition, CI_95_) by the method of Chou and Talalay [24], using CalcuSyn software version 1.0 (Biosoft, Cambridge, United Kingdom). Weighted average CI values (CI_wt_) were calculated as (CI_50_ + 2 × CI_75_ + 3 × CI_90_ + 4 × CI_95_)/10. CI_wt_ values ≤0.7 indicate synergistic effects, CI_wt_ values >0.7 and ≤0.9 moderately synergistic effects, CI_wt_ values >0.9 and ≤1.2 additive effects, CI_wt_ values >1.2 and ≤1.45 moderately antagonistic effects, and CI_wt_ values >1.45 antagonistic effects [24].

### 2.5. Viability Assay

Cell viability was assessed using the 3-(4,5-dimethylthiazol-2-yl)-2,5-diphenyltetrazolium bromide (MTT) dye reduction assay as described previously [23]. Five thousand cells were seeded per well in 96-well microtiter plates and incubated with culture medium containing serial dilutions of the indicated substances. After five days of incubation, MTT (1 mg/mL) was added, and after an additional 4 h, cells were lysed in a buffer containing 20% (*w/v*) sodium dodecyl sulfate (SDS) and 50% *N*,*N*-dimethylformamide adjusted to pH 4.5. Absorbance was determined at 570 nm for each well using a 96-well multiscanner. After subtracting background absorbance, cell viability was expressed in percent relative to untreated control cells. Drug concentrations that reduced cell viability by 50% (CC_50_) were calculated using CalcuSyn (Biosoft, Cambridge, UK). The MTT assay measures metabolic activity in the mitochondria. To confirm viability results by a second assay, the CellTiter-Glo assay (Promega, Walldorf, Germany), which measures cellular ATP production, was used according to the manufacturer’s instructions.

### 2.6. Virus Yield Assay

The amount of infectious virus was determined by virus yield assay in a single-cycle assay format, as previously described [23]. Virus titres were expressed as 50% of tissue culture infectious dose (TCID_50_/mL), 120 h post infection.

### 2.7. Immunoblotting

Immunoblotting was performed as previously described [23]. In brief, cells were lysed in a Triton X-100 sample buffer and proteins separated by SDS-polyacrylamide gel electrophoresis (PAGE). Proteins were detected using specific antibodies against ß-actin (3598R-100-BV, BioVision via BioCat, Heidelberg, Germany) or HCMV 45 kDa LA (MBS320051, MyBioSource via Biozol, Echingen, Germany), and were visualized by enhanced chemiluminescence using a commercially available kit (Thermo Scientific, Schwerte, Germany).

### 2.8. Statistics

Values presented are the mean ± S.D. of three independent biological repeats. Comparisons between two groups were performed using the Student’s *t*-test, three and more groups were compared by ANOVA, followed by the Student–Newman–Keuls test. Data groups were considered significantly different at *p* < 0.05.

## 3. Results

### 3.1. Eltrombopag Inhibits HCMV Replication in Human Foreskin Fibroblasts by Interference with Late Processes of the Replication Cycle

Eltrombopag did not affect HCMV Hi91-induced immediate early antigen (IEA) expression (Appendix A), but inhibited HCMV Hi91-induced LA expression with an IC_50_ of 415 nM in HFFs (Figure 1A,B). Eltrombopag concentrations of up to 25 µM did not reduce the viability of proliferating HFFs by 50%, as determined by MTT assay (Figure 1A). Cell viability determination by CellTiter-Glo resulted in similar results (HFF viability at 25 µM: 53 ± 4 µM). Hence, the selectivity index, cytotoxicity concentration (CC_50_)/IC_50,_ is higher than 60.2 (Figure 1A). Higher MOIs were associated with higher IC_50_ values (Figure 1C). At MOI 1, the highest MOI investigated in HFFs, the eltrombopag IC_50_ was 3844 nM. The observed eltrombopag concentrations are within the range of therapeutic plasma concentrations, which have been described to exceed 45 µM [25,26].

Eltrombopag-induced inhibition of HCMV LA translated into reduced virus replication as indicated by virus yield assay (Figure 2A). At a concentration of 10 µM, eltrombopag reduced virus titres by 1.8 × 10^4^-fold and at 500 nM still by 15-fold.

The HCMV replication cycle is divided into three phases characterized by the expression of immediate early, early, and late viral genes. Immediate early genes are transcribed immediately after infection and do not depend on synthesis of viral DNA or transcription of proteins. Delayed early proteins are represented by the viral DNA polymerase, and other viral functions required for viral DNA synthesis, and some viral structural proteins. Late genes encode mostly structural proteins used in viral assembly and packaging, and are generally expressed subsequent to delayed early genes [27].

To better define which phases of the viral replication cycle are affected by eltrombopag, the drug was added at different time points (Figure 2B, Appendix A). Pre-incubation and drug addition during the 1-h virus adsorption period did not, or only modestly, affect virus replication. This shows that eltrombopag does not primarily interfere with virus binding to host cells and virus internalization, but needs to be present during virus replication to exert its anti-HCMV effects. Drug addition 1-h or 24 h post infection was sufficient to achieve maximum inhibition of HCMV LA expression (Figure 2B, Appendix A). This, together with the observed lack of inhibition of HCMV immediate early antigen (IEA) expression, indicates that eltrombopag inhibits the late stages of the HCMV replication cycle characterized by LA expression. Drug addition 48 h post infection resulted in reduced effects compared to drug addition 1-h or 24 h post infection (Figure 2B, Appendix A).

### 3.2. Eltrombopag Inhibits HCMV Expression via Iron Chelation

Eltrombopag was developed as a thrombopoietin receptor agonist [1,2,3]. However, it is unlikely that eltrombopag inhibits HCMV replication via thrombopoietin receptor activation, because fibroblasts do not express the thrombopoietin receptor [28]. In agreement, eltrombopag also inhibited murine cytomegalovirus replication in murine NIH/3T3 fibroblasts (Figure 3A), although eltrombopag does not target the murine thrombopoietin receptor [29].

Eltrombopag is also an iron chelator [2,30,31], and iron chelators have been shown to inhibit HCMV replication [32,33,34,35,36,37,38]. The addition of equimolar Fe^3+^ concentrations was shown to inhibit pharmacological action of eltrombopag that are caused via iron chelation [31]. Hence, we investigated eltrombopag in combination with equimolar Fe(III)Cl_3_ concentrations to investigate whether iron chelation is the mechanism by which eltrombopag exerts its anti-HCMV effects (Figure 3B). Equimolar Fe(III)Cl_3_ concentrations prevented the anti-HCMV effects of eltrombopag (Figure 3B). This suggests that iron chelation is the main mechanism of eltrombopag’s anti-HCMV activity. In agreement, eltrombopag exerted antagonistic effects in combination with the iron chelator deferasirox (Appendix A), which may indicate that both compounds share the same antiviral mechanism.

### 3.3. Eltrombopag Exerts Synergistic Effects with Ganciclovir

Next, we tested eltrombopag in combination with ganciclovir, the mainstay of anti-HCMV therapies [13]. The combination of equimolar eltrombopag and ganciclovir concentrations resulted in synergistic anti-HCMV effects (Figure 4), which is illustrated by a weighted average combination index (CI_WT_) of 0.17 ± 0.03, as determined by the method of Chou and Talalay [24]. According to this method, combined effects are considered to be synergistic at a CI_WT_ of ≤0.7 [24]. The combination of eltrombopag and foscarnet also displayed synergistic effects (Appendix A).

### 3.4. Eltrombopag Is Effective in Different Cell Types and against Different Virus Strains and Isolates Including Drug-Resistant Ones

Finally, we investigated the effects of eltrombopag against a broader range of laboratory virus strains and clinical isolates in HFFs, and primary-adipose-derived ASCs, another cell type that supports HCMV replication [39]. The laboratory HCMV strains included Davis [40] and Towne [41] in addition to Hi91. The clinical isolates U1, U59, and U75 were isolated from the urine of patients, as previously described [20,21]. U1 and U59 harbor an A987G mutation in the HCMV DNA polymerase UL54 (Table 1), which is known to confer combined ganciclovir and cidofovir resistance [42,43]. U1 also displays a C607Y mutation in the HCMV kinase UL97 (Table 1), which is associated with ganciclovir resistance [44,45]. In agreement, U1 and U59 were characterized by high ganciclovir and cidofovir IC_50_s (Table 1), which are typically considered to indicate resistance [46,47,48]. U75 also displayed resistance to ganciclovir and cidofovir, although it does not harbor known resistance mutations (data not shown).

The eltrombopag IC_50_s ranged from 99 nM (U1 in HFFs) to 4331 nM (Hi91 in ASCs) (Figure 5A, Appendix A). When compared across the two cell types, the different HCMV strains and clinical isolates displayed similar eltrombopag sensitivity, apart from U1, which appeared to be particularly sensitive to eltrombopag in HFFs and ASCs (Figure 5A). The average HCMV sensitivity to eltrombopag was very similar in both cell types.

To confirm the relevance of iron chelation as mechanism of the anti-HCMV action of eltrombopag using a clinical virus isolate, U1-infected HFFs were treated with equimolar concentrations of eltrombopag and Fe(III)Cl_3_. The presence of equimolar Fe^3+^ concentrations prevented the eltrombopag-induced inhibition of HCMV LA expression in U1-infected cells in a comparable fashion (Figure 5B), as in Hi91-infected cells (Figure 3B).

## 4. Discussion

Here, we show that the approved thrombopoietin receptor agonist eltrombopag exerts anti-HCMV effects in various cell types infected with a range of different virus strains and clinical isolates, including drug-resistant ones. The observed IC_50_ values ranged from 99 nM to 4331 nM, which is in the range of therapeutic plasma concentrations that have been reported to exceed 45 µM [25,26]. Eltrombopag also synergistically increased the activity of the approved anti-HCMV drug ganciclovir.

Our findings are in agreement with a case report on an immunocompetent patient, who suffered from HCMV-associated thrombocytopenia and recovered after eltrombopag therapy [17]. This response had originally been attributed to effects of eltrombopag on platelet production [17]. The possibility that eltrombopag may exert antiviral affects was not considered. Our current data show that therapeutic eltrombopag levels interfere with HCMV replication, which may have contributed to the beneficial clinical outcome. Notably, eltrombopag has also been shown to inhibit the replication of severe fever with thrombocytopenia syndrome virus, a member of the genus *Banyangvirus* (Phenuiviridae) [49].

The anti-HCMV effects of eltrombopag are unlikely to be caused by action on the thrombopoietin receptor, since eltrombopag was effective in cell types that do not express the thrombopoietin receptor, which is expressed in hematopoietic cells [28,29]. In agreement, eltrombopag also exerted antiviral effects in mouse fibroblasts infected with murine CMV, although the hematological effects of eltrombopag are known to be species-specific and to not affect mice [28,29].

Eltrombopag is also known to be an iron chelator [30,31]. The addition of Fe^3+^ prevented the eltrombopag-mediated anti-HCMV effects in strain Hi91 and clinical isolate U1-infected cells. Hence, our data suggest that eltrombopag inhibits HCMV replication via Fe^3+^ chelation.

A number of different iron chelators, including desferrioxamine, diethylenetriaminepentaacetic acid (DTPA), and ethylenediaminedisuccinic acid (EDDS), were shown to inhibit HCMV replication [32,33,34,35,36,37,38]. However, the iron chelators tiron and ciclopirox olamine were not found to inhibit HCMV strain AD169 replication in MRC5 cells [50]. The experimental setup differed, as MRC5 cells were infected at a high MOI of 3, and no dose–response relationships were determined. Hence, a direct comparison is not possible. Notably, specific antiviral activity can easily be missed if the therapeutic window between antiviral and cytotoxic effects is relatively small. For example, desferrioxamine was found to inhibit HCMV replication at concentrations that did not decrease the viability of confluent fibroblasts but affected dividing cells [32]. In contrast, eltrombopag inhibits HCMV replication in concentrations that do not affect cell proliferation. Hence, the size of the therapeutic window that discriminates between anti-HCMV activity and antiproliferative and cytotoxic effects substantially differs among iron chelators. Eltrombopag seems to be an iron chelator that possesses a particularly preferential therapeutic window in terms of its anti-HCMV activity.

Due to its effects on platelet counts and hematopoietic stem cells [2], however, the anti-HCMV effects of eltrombopag are primarily of relevance for anemia patients at risk of HCMV disease, for whom eltrombopag is indicated. Eltrombopag has been suggested for the treatment of cytopenias after hematopoietic stem cell transplantations and case reports support its safety and efficacy [4,5,6,7,8,9]. Since HCMV reactivation and HCMV-associated diseases are leading reasons for the failure of hematopoietic stem cell transplantations [10,11,12], antiviral effects exerted by eltrombopag may also contribute to improved therapy outcome. Notably, eltrombopag was effective against resistant clinical HCMV isolates, and resistance formation to the approved drugs is a major challenge after stem cell transplantation [11,12].

## 5. Conclusions

Therapeutic eltrombopag concentrations inhibit HCMV replication via chelation of Fe^3+^ ions. Eltrombopag is effective against drug-resistant viruses and synergistically increases the effects of the mainstay anti-HCMV drug ganciclovir. The anti-HCMV activity of eltrombopag may be of particular interest for its use for the treatment of cytopenias after haematopoietic stem cell transplantation, as HCMV reactivation and disease is a major reason for transplantation failure.

## Figures and Tables

**Figure 1 cells-09-00031-f001:**
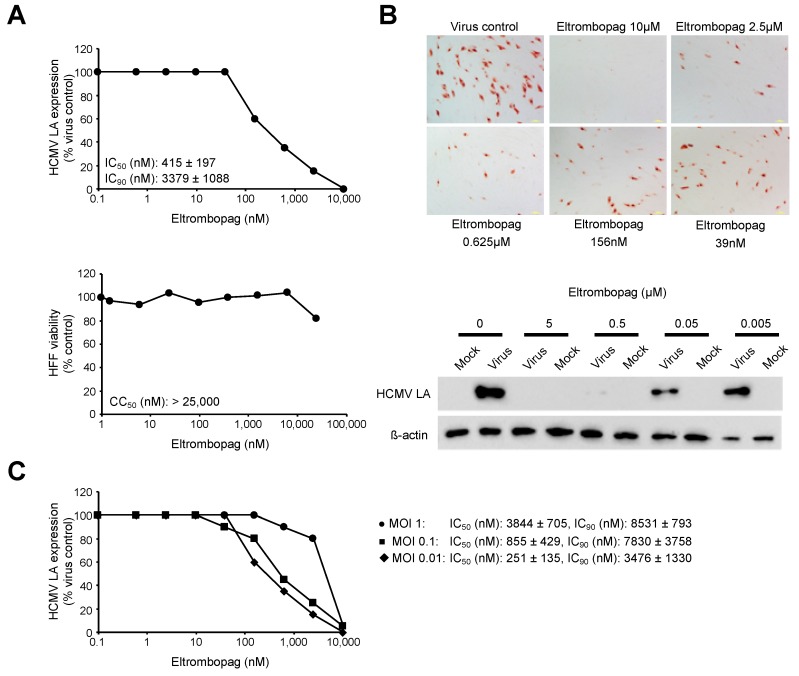
Effects of eltrombopag on human cytomegalovirus (HCMV) late antigen (LA) expression in primary human foreskin fibroblasts (HFFs). (**A**) Representative dose–response curves showing the effects of eltrombopag on HCMV LA expression and HFF viability (as determined after 120 h of incubation). Eltrombopag concentrations that reduce HCMV LA expression by 50% (inhibitory concentration (IC)_50_) or 90% (IC_90_) and cell viability by 50% (cytotoxicity concentration (CC)_50_) relative to untreated controls are also provided. Eltrombopag was continuously present from the time of virus infection. (**B**) Representative photographs and Western blots demonstrating the effects of eltrombopag on HCMV LA expression. In (**A**,**B**), HFFs were infected with HCMV strain Hi91 (multiplicity of infection (MOI) 0.02). HCMV LA expression was detected 120 h post infection. (**C**) Representative dose–response curves and IC_50_ values indicating effects of eltrombopag on HCMV LA expression in HFFs infected with different MOIs of HCMV strain Hi91 as detected 120 h post infection.

**Figure 2 cells-09-00031-f002:**
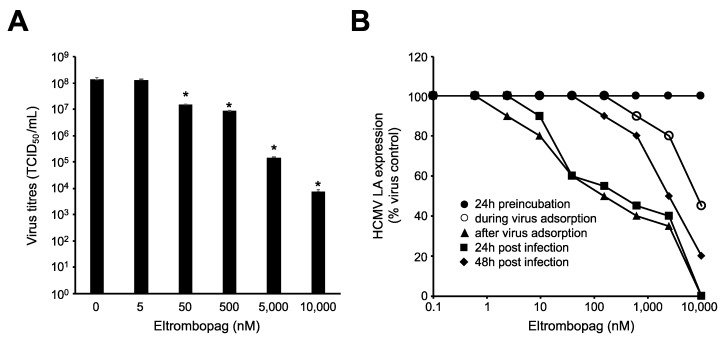
Effects of eltrombopag on HCMV replication and at different stages of the viral replication cycle. HFFs were infected with HCMV strain Hi91 (MOI 0.02). HCMV LA expression and virus titres were detected 120 h post infection. (**A**) Virus titres in the absence or presence of eltrombopag. (**B**) Representative dose–response curves and IC_50_ values indicating the effects of eltrombopag on HCMV LA expression after 24 h of pre-treatment, after treatment during the 1-h adsorption period, after drug addition post infection following the 1-h virus adsorption period, after drug addition 24 h post infection, and after drug addition 48 h post infection. * *p* < 0.05

**Figure 3 cells-09-00031-f003:**
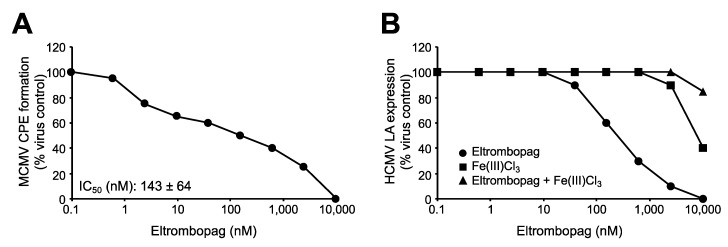
Eltrombopag inhibits HCMV infection by iron depletion. (**A**) Representative dose–response curve indicating the effects of eltrombopag on cytopathogenic effect (CPE) formation (detected 120 h post infection), in murine cytomegalovirus (MOI 1)-infected murine, NIH/3T3 fibroblasts, and eltrombopag concentration that reduces CPE formation by 50% (IC_50_), relative to untreated control. The findings indicate that eltrombopag interferes with cytomegalovirus replication by thrombopoietin-receptor-independent effects, since eltrombopag does not activate the murine thrombopoietin receptor. The investigated eltrombopag concentrations did not affect NIH/3T3 cell viability. (**B**) Representative growth curve indicating the effects of equimolar concentrations of Fe(III)Cl_3_ on the anti-HCMV effects of eltrombopag, as indicated by HCMV LA expression in HCMV Hi91 (MOI 0.02)-infected human foreskin fibroblasts (HFFs) 120 h post infection. Equimolar Fe(III)Cl_3_ concentrations circumvent the anti-HCMV effects exerted by eltrombopag.

**Figure 4 cells-09-00031-f004:**
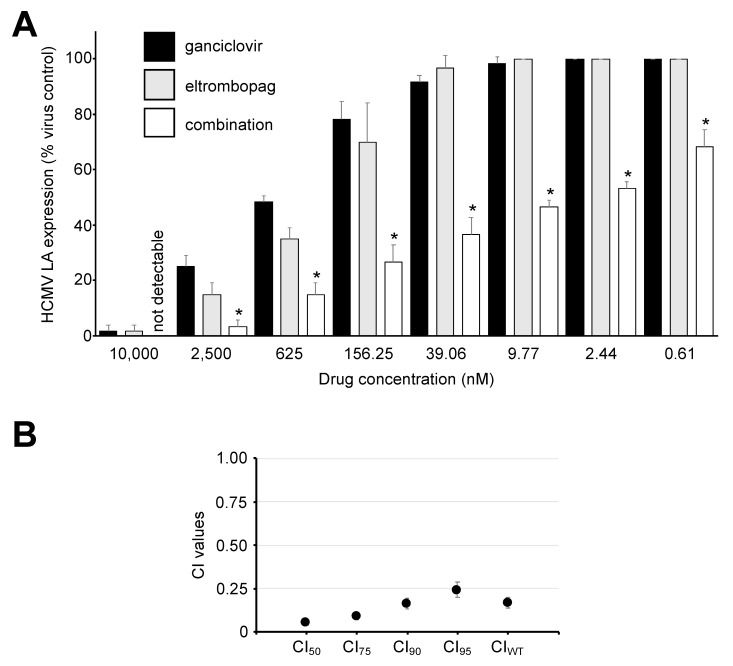
Antiviral effects of eltrombopag in combination with ganciclovir. (**A**) Effects of equimolar drug concentrations on HCMV LA expression in HCMV Hi91 (MOI 0.02) HFFs 120 h post infection. * *p* < 0.05 compared to either single treatment; (**B**) Combination indices (CIs) at different levels of inhibition and weighted average CI values (CI_wt_) calculated as (CI_50_ + 2 × CI_75_ + 3 × CI_90_ + 4 × CI_95_)/10 [24]. CI_wt_ values ≤0.7 indicate synergistic effects [24].

**Figure 5 cells-09-00031-f005:**
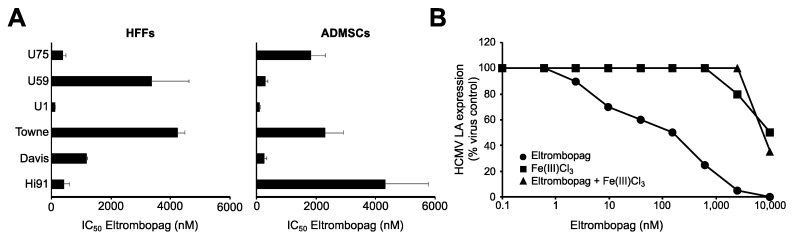
Antiviral effects of eltrombopag determined in different cell types infected by different HCMV strains and clinical isolates. HFFs were infected at an MOI of 0.02 and adipose-derived adult mesenchymal stem cells (ASCs) at an MOI of 5. HCMV LA expression was determined 120 h post infection. (**A**) Eltrombopag concentrations that reduce HCMV LA expression by 50% (IC_50_). Numerical values are provided in Appendix A. The eltrombopag concentration that reduced ASC viability by 50% (CC_50_) was 17,872 ± 1302 nM. (**B**) Representative growth curve indicating the effects of equimolar concentrations of Fe(III)Cl_3_ on the anti-HCMV effects of eltrombopag as indicated by HCMV LA expression in U1 (MOI 0.02)-infected HFFs 120 h post infection.

**Table 1 cells-09-00031-t001:** Sensitivity of laboratory HCMV strains and patient isolates to ganciclovir and cidofovir as indicated by HCMV LA expression in human foreskin fibroblasts infected at MOI 0.02 120 h post infection. Concentrations that reduce LA expression by 50% (IC_50_) are provided.

		IC_50_ (µM)	IC_50_ (µM)
Virus Strain/Isolate	Resistance Mutations	Ganciclovir	Cidofovir
Hi91	n/a	0.71 ± 0.57	0.31 ± 0.14
Davis	n/a	0.72 ± 0.31	0.17 ± 0.02
Towne	n/a	0.61 ± 0.12	0.14 ± 0.01
U1	A987G *, C607Y **	24 ± 10	5.2 ± 1.9
U59	A987G	23 ± 14	1.6 ± 0.5

* mutation in the HCMV DNA polymerase UL54, which confers resistance to ganciclovir and cidofovir [42,43]. ** mutation in the HCMV kinase UL97, which is associated with ganciclovir resistance [44,45].

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
