# Peer review of "The Thrombopoietin Receptor Agonist Eltrombopag Inhibits Human Cytomegalovirus Replication Via Iron Chelation"

_cells, 2019, doi:10.3390/cells9010031_

Round 1

Reviewer 1 Report

In “The thrombopoietin receptor agonist eltrombopag 3 inhibits human cytomegalovirus replication via iron 4 chelation” Vogel, et al describe inhibition of human cytomegalovirus (HCMV) replication in cell culture models. The authors put forth the idea eltrombopag, which is traditionally used as a pro-platelet drug that acts through thrombopoietin-like signaling, blocks HCMV replication by chelating iron. The current works builds upon several previous studies that demonstrate that the replication of herpesviruses, among other viruses, is blocked by iron chelating agents. The authors use two cell types and several HCMV strains to show that eltrombopag limits HCMV replication. The manuscript describes work that adds to the list of agents that may block HCMV replication by chelating metals. The presented data is compelling and the writing mostly clear. The following are suggestions for improving the manuscript.

There are two concerns regarding the cytotoxicity data presented. First, cytotoxicity was measured using MTT. MTT assays measurements depend on the cell’s metabolic status, which could be altered by iron chelation. The authors should consider confirming the cytotoxicity of eltrombopag using an assay that is independent on the intracellular metabolic environment. It is unclear if the authors evaluated cytotoxicity in ASCs. That must be shown in addition to the HFF results. Second, the authors state that “eltrombopag inhibits HCMV replication in concentrations that do not affect cell proliferation” (line 286). However, no proliferation data is presented. Presumably the authors are making such a statement since the MTT assay was performed on cells that were in a proliferative state when the assay as initiated. The authors should consider showing proliferation data or more carefully word their descriptions.

The authors place a lot of attention on the possible clinical benefit of treating an HCMV-infected person with eltrombopag and cite a previous study involving a patient with a low platelet count (Simpson, et al. Intern Med J. 2016). However, the clinical potential of eltrombopag as a treatment for HCMV disease would be limit by the potentially harmful effects of raising platelet levels in a person that would otherwise have a normal count prior to treatment. Since the authors place much emphasis on the clinical antiviral potential of eltrombopag they should address this limitation in their discussion.

The authors show that eltrombopag and ganicicolovir treatment together provide a greater loss in replication than treatment with either agent alone. This data is included to further support the clinical potential of eltrombopag. The observation also suggest that each agent inhibits HCMV through independent mechanisms. A more interesting experiment in terms of understanding eltrombopag mechanism of action against HCMV would the use of eltrombopag and a known iron chelator that is characterized to limit HCMV replication. Presumably, the two chelators would not act synergistically, and iron addition would equally rescue. Such a finding would support the author’s hypothesis that eltrombopag’s antiviral mechanism of action is through iron chelation. If the authors would find that two chelators act synergistically, then it would suggest that eltrombopag has an additional or alternative mechanism of action.

In Figure 2, 108/mL is a high TCID50 for HCMV. Does this value represent infectious virus released by cells or does this also include cell-associated virus?

In Figure 5, should the eltrombopag concentrations be listed in nM or is µM correct? If µM is correct, these findings would be inconsistent with the other figures and with the descriptions provided in the text.

Figure 5B and C seem unnecessary. It is unclear why the authors would avg the cell types / virus strains. Figure 5A should suffice and provide the best clarity of the observed results.

Grammatical issues:

Line 109: “per cent” needs to be one word

Line 298: “Ttherapeutic” needs to be changed to Therapeutic

Author Response

In “The thrombopoietin receptor agonist eltrombopag 3 inhibits human cytomegalovirus replication via iron 4 chelation” Vogel, et al describe inhibition of human cytomegalovirus (HCMV) replication in cell culture models. The authors put forth the idea eltrombopag, which is traditionally used as a pro-platelet drug that acts through thrombopoietin-like signaling, blocks HCMV replication by chelating iron. The current works builds upon several previous studies that demonstrate that the replication of herpesviruses, among other viruses, is blocked by iron chelating agents. The authors use two cell types and several HCMV strains to show that eltrombopag limits HCMV replication. The manuscript describes work that adds to the list of agents that may block HCMV replication by chelating metals. The presented data is compelling and the writing mostly clear. The following are suggestions for improving the manuscript.

There are two concerns regarding the cytotoxicity data presented. First, cytotoxicity was measured using MTT. MTT assays measurements depend on the cell’s metabolic status, which could be altered by iron chelation. The authors should consider confirming the cytotoxicity of eltrombopag using an assay that is independent on the intracellular metabolic environment.

Authors’ response:

The results were confirmed using the CellTiterGlo assay, which uses cellular ATP production as surrogate for the determination of cell viability. Section 2.5 was amended as follows to indicate this (p. 3, lines 95-97):

"The MTT assay measures metabolic activity in the mitochondria. To confirm viability results by a second assay, the CellTiterGlo assay (Promega, Walldorf, Germany), which measures cellular ATP production, was used according to the manufacturer's instructions."

Section 3.1 was amended as follows (p. 4, lines 141-143):

"Eltrombopag concentrations of up to 25µM did not reduce the viability of proliferating HFFs by 50%, as determined by MTT assay (Figure 1A). Cell viability determination by CellTiterGlo resulted in similar results (HFF viability at 25µM: 53 ± 4µM)."

It is unclear if the authors evaluated cytotoxicity in ASCs. That must be shown in addition to the HFF results. Second, the authors state that “eltrombopag inhibits HCMV replication in concentrations that do not affect cell proliferation” (line 286). However, no proliferation data is presented. Presumably the authors are making such a statement since the MTT assay was performed on cells that were in a proliferative state when the assay as initiated. The authors should consider showing proliferation data or more carefully word their descriptions.

Authors’ response:

The CC50 value for eltrombopag is 17.8 ± 1.3µM. This information was added to the legend of Figure 5 (p. 8, lines 266-267) and Table S3. The MTT was indeed performed on proliferating cells, since proliferating cells are more sensitive to drug effects than confluent cells. Based on this, we feel that we can conclude that eltrombopag inhibits HCMV replication in concentrations that did not interfere with cell proliferation. This is relevant, because it is in contrast to other iron chelators that only inhibit virus replication at concentrations that, although they do not affect the viability of cells in monolayers, inhibit cell proliferation.

The authors place a lot of attention on the possible clinical benefit of treating an HCMV-infected person with eltrombopag and cite a previous study involving a patient with a low platelet count (Simpson, et al. Intern Med J. 2016). However, the clinical potential of eltrombopag as a treatment for HCMV disease would be limit by the potentially harmful effects of raising platelet levels in a person that would otherwise have a normal count prior to treatment. Since the authors place much emphasis on the clinical antiviral potential of eltrombopag they should address this limitation in their discussion.

Authors’ response:

Many thanks for this comment. We did not mean to suggest the use of eltrombopag as general antiviral drug. To clarify that the anti-HCMV effects of eltrombopag are specifically of relevance for patients at risk of HCMV disease, for whom eltrombopag is indicated due to anaemic conditions, the last paragraph of the Discussion was amended as follows (p. 9, lines 306-308):

“Due to its effects on platelet counts and haematopoietic stem cells [2], however, the anti-HCMV effects of eltrombopag are primarily of relevance for anaemia patients at risk of HCMV disease for whom eltrombopag is indicated.”

The authors show that eltrombopag and ganicicolovir treatment together provide a greater loss in replication than treatment with either agent alone. This data is included to further support the clinical potential of eltrombopag. The observation also suggest that each agent inhibits HCMV through independent mechanisms. A more interesting experiment in terms of understanding eltrombopag mechanism of action against HCMV would the use of eltrombopag and a known iron chelator that is characterized to limit HCMV replication. Presumably, the two chelators would not act synergistically, and iron addition would equally rescue. Such a finding would support the author’s hypothesis that eltrombopag’s antiviral mechanism of action is through iron chelation. If the authors would find that two chelators act synergistically, then it would suggest that eltrombopag has an additional or alternative mechanism of action.

Authors’ response:

We combined eltrombopag with the iron chelator deferasirox, and saw an antagonism, which supports that deferasirox and eltrombopag do not inhibit HCMV replication by different mechanisms. The data are presented in the new Table S2. Section 3.2 was amended as follows (p. 6, lines 214-216):

“In agreement, eltrombopag exerted antagonistic effects in combination with the iron chelator deferasirox (Table S2), which may indicate that both compounds share the same antiviral mechanism.”

In Figure 2, 108/mL is a high TCID50 for HCMV. Does this value represent infectious virus released by cells or does this also include cell-associated virus?

Authors’ response:

The virus titres were determined in cell culture supernatants. Hence, this number refers to viruses that had been released by the cells.

In Figure 5, should the eltrombopag concentrations be listed in nM or is µM correct? If µM is correct, these findings would be inconsistent with the other figures and with the descriptions provided in the text.

Authors’ response:

Many thanks, yes, it should be nM. We corrected this.

Figure 5B and C seem unnecessary. It is unclear why the authors would avg the cell types / virus strains. Figure 5A should suffice and provide the best clarity of the observed results.

Authors’ response:

We removed Figure 5B and C.

Grammatical issues:

Line 109: “per cent” needs to be one word

Line 298: “Ttherapeutic” needs to be changed to Therapeutic

Authors’ response:

This was corrected.

Reviewer 2 Report

Thank you for giving me the opportunity to review this interesting work. The authors described that the efficacy of eltrombopag on CMV-induced thrombopenia seems to be partialy due to direct antiviral effects via iron chelation.

This article is well written; the conclusions are supported by the in vitro results and well discussed in the Discussion part. More, if those will be confirmed in future clinical studies, eltrombopag could be an interesting candidate to treat CMV infections/diseases in thrombopenic patients. It could be of major interest in hematopoietic stem cell transplant récipients, as the use of the "gold standard" to treat CMV infections/diseases in this population (valganciclovir/ganciclovir) is limited by its frequent hematotoxicity.

The authors described the synergistic effects between eltrombopag and ganciclovir. We suggest to study also the effects between eltrombopag and other major antivirals known to be effective against CMV (foscarnet, letermovir, maribavir).

Author Response

Thank you for giving me the opportunity to review this interesting work. The authors described that the efficacy of eltrombopag on CMV-induced thrombopenia seems to be partialy due to direct antiviral effects via iron chelation.

This article is well written; the conclusions are supported by the in vitro results and well discussed in the Discussion part. More, if those will be confirmed in future clinical studies, eltrombopag could be an interesting candidate to treat CMV infections/diseases in thrombopenic patients. It could be of major interest in hematopoietic stem cell transplant récipients, as the use of the "gold standard" to treat CMV infections/diseases in this population (valganciclovir/ganciclovir) is limited by its frequent hematotoxicity.

The authors described the synergistic effects between eltrombopag and ganciclovir. We suggest to study also the effects between eltrombopag and other major antivirals known to be effective against CMV (foscarnet, letermovir, maribavir).

Authors’ response:

We also combined eltrombopag with foscarnet and found synergistic effects. The results are presented in the new Table S2. Section 3.3 was amended as follows (p. 6, lines 222-223):

“The combination of eltrombopag and foscarnet also displayed synergistic effects (Table S2).”

Reviewer 3 Report

Remarks to the Author:

This work by Vogel and colleagues describes studies examining the anti-HCMV potency of thrombopoietin receptor agonist eltrombopag in human fibroblasts and adult mesenchymal stem cells.

The central message of the paper is that eltrombopag exerts antiviral activity via Fe3+ chelation. Furthermore, it synergistically increases the effect of gancyclovir and it’s active against drug resistent HCMV. The paper clarifies the antiviral mechanism of action with well-established experimental procedures in the field of virology.

The topic meets the scope of Cells but several revisions are required to be suitable for publication.

Revisions:

Introduction.

Please mention in the text the use of eltrombopag in patients with hepatitis C virus-associated thrombocytopenia and relative references, like Elbedewy et al, Ther Clin Risk Manag. 2019.

Materials and Methods.

There are no indications on the viral titration in the paragraph 2.2. Please add them. Add information about the number of sedded cells in cellular assays Give more details about the virus yield assay and immunoblotting. What is the experimental protocol (cells, MOI, etc.)? In virus infectivity assay the description of “MCMV infectivity assay” is missing.  

Results.

1. Please provide the IC90 of eltrombopag Figure 1 would be more complete adding data on IEA expression (mentioned in the text) in the panel 1A and in the western blotting. Figure 1B: order the pictures according to the increasing dose 1 and 2. Graphs of dose response curves are more explicative expressing the concentration as Log (mM). Add the standard deviations in graphs. Line 202: Change “concluded” to “supposed” Line 218: it’s not clear if ASCs have the thrombopoietin receptor. Table 1. Insert IC50 and IC90 and SI values of Eltrombopag  against viruses. It will be interesting to evaluate the antiviral activity also against the vascular endotheliotropic strain TB40/E. Figure 5 B and C. I don’t agree to show the average of IC50 in HFFs/ASCs since two different MOI were used in these cells. 5D: Has the effect of eltrombopag on IEA expression of these viruses been demonstrated?

Discussion.

Line 267: Being the thrombocytopenia syndrome virus the only other virus inhibited by eltrombopag according to literature, please give major details on this antiviral activity and mechanism of action.

Supplementary Materials.

Insert results on cellular viability that are mentioned in the manuscript.

Author Response

This work by Vogel and colleagues describes studies examining the anti-HCMV potency of thrombopoietin receptor agonist eltrombopag in human fibroblasts and adult mesenchymal stem cells.

The central message of the paper is that eltrombopag exerts antiviral activity via Fe3+ chelation. Furthermore, it synergistically increases the effect of gancyclovir and it’s active against drug resistent HCMV. The paper clarifies the antiviral mechanism of action with well-established experimental procedures in the field of virology.

The topic meets the scope of Cells but several revisions are required to be suitable for publication.

Revisions:

Introduction.

Please mention in the text the use of eltrombopag in patients with hepatitis C virus-associated thrombocytopenia and relative references, like Elbedewy et al, Ther Clin Risk Manag. 2019.

Authors’ response:

This was done. The first sentence of the Introduction was amended as follows (p. 1, lines 41-42):

“Eltrombopag is a thrombopoietin receptor (also known as c‐Mpl or MPL) agonist that is used for the treatment of thrombocytopenia including hepatitis C virus-associated thrombocytopenia [1-3].”

The use of eltrombopag against hepatitis C virus-associated thrombocytopenia is covered by reference 3.

Materials and Methods.

There are no indications on the viral titration in the paragraph 2.2. Please add them. Add information about the number of sedded cells in cellular assays Give more details about the virus yield assay and immunoblotting. What is the experimental protocol (cells, MOI, etc.)? In virus infectivity assay the description of “MCMV infectivity assay” is missing.

Authors’ response:

This was corrected. Since different MOIs were used for different cell/ virus combinations the MOIs for individual experiments are provided in the respective figure legends. The legend of Table 1 was amended as follows (p. 8, lines 249-251):

“Table 1. Sensitivity of laboratory HCMV strains and patient isolates to ganciclovir and cidofovir as indicated by HCMV late antigen (LA) expression in human foreskin fibroblasts infected at MOI 0.02 120h post infection. Concentrations that reduce LA expression by 50% (IC50) are provided.”

Section 2.2 was amended as follows to provide information on the cultivation of MCMV (p. 2, lines 77-79):

“Murine cytomegalovirus (Smith strain, catalogue number VR-1399) was obtained from ATCC and virus stocks were prepared in NIH/3T3 mouse fibroblasts (ATCC) maintained in MEM supplemented with 4% FCS.”

Section 2.3 ‘Virus infectivity assay’ was amended as follows (p. 3, lines 95-97):

“Effects of eltrombopag on murine cyromegalovirus were determined by visual scoring of cytopathogenic effect (CPE) formation (detected 120h post infection) in MOI 1-infected murine NIH/3T3 fibroblasts.”

A link to the method used for virus titration is provided in section 2.6.

Section 2.5 ‘Viability assay’ was amended as follows (p. 3, lines 109-111):

“5,000 cells were seeded per well in 96-well microtiter plates and incubated with culture medium containing serial dilutions of the indicated substances.”

Results.

Please provide the IC90 of eltrombopag Figure 1 would be more complete adding data on IEA expression (mentioned in the text) in the panel 1A and in the western blotting.

Authors’ response:

The IC90 values were added to Figure 1. IEA expression was measured by immunostaining. The data was added as new Figure S1.

Figure 1B: order the pictures according to the increasing dose 1 and 2. Graphs of dose response curves are more explicative expressing the concentration as Log (mM). Add the standard deviations in graphs.

Authors’ response:

We are afraid, we do not understand the criticism about the axes. They are on a logarithmic scale. Figure 1B does not include any graphs.

Line 202: Change “concluded” to “supposed” Line 218: it’s not clear if ASCs have the thrombopoietin receptor.

Authors’ response:

The sentence was rephrased accordingly as follows (p. 6, lines 213-214):

“Equimolar Fe(III)Cl3 concentrations prevented the anti-HCMV effects of eltrombopag (Figure 3B). This suggests that iron chelation is the main mechanism of eltrombopag’s anti-HCMV activity.”

We are not sure what the second comment refers to.

Table 1. Insert IC50 and IC90 and SI values of Eltrombopag  against viruses. It will be interesting to evaluate the antiviral activity also against the vascular endotheliotropic strain TB40/E.

Authors’ response:

The respective data are provided in Table S3. Although we agree that it would be interesting to include additional HCMV strains and clinical isolates, we think that three laboratory strains and three clinical isolates should be sufficient to enable solid conclusions.

Figure 5 B and C. I don’t agree to show the average of IC50 in HFFs/ASCs since two different MOI were used in these cells.

Authors’ response:

We removed them.

5D: Has the effect of eltrombopag on IEA expression of these viruses been demonstrated?

Authors’ response:

No. Since we did not detect any effect of eltrombopag on IEA expression in Hi91-infected HFFs, we did not investigate IEA expression in other HCMV-infected cell cultures.

Discussion.

Line 267: Being the thrombocytopenia syndrome virus the only other virus inhibited by eltrombopag according to literature, please give major details on this antiviral activity and mechanism of action.

Authors’ response:

Eltrombopag was identified in a screen of an FDA-approved drug library for activity against thrombocytopenia syndrome virus. It reduced the production of the viral nucleoprotein NP, as determined by ELISA, and of the S segment, as determined by qPCR. The study then focused on another drug candidate. Hence, there is no in-depth mechanistic information available, which could be discussed.

Supplementary Materials.

Insert results on cellular viability that are mentioned in the manuscript. 

Authors’ response:

The CC50 for eltrombopag in ASCs was added to the legend of Figure 5 (p. 8, lines 266-267) and Table S3.

Round 2

Reviewer 1 Report

The authors responses to the previous comments are appropriate.